# Perinatal Distress and Depression in Culturally and Linguistically Diverse (CALD) Australian Women: The Role of Psychosocial and Obstetric Factors

**DOI:** 10.3390/ijerph16162945

**Published:** 2019-08-16

**Authors:** Felix Akpojene Ogbo, Osita Kingsley Ezeh, Mansi Vijaybhai Dhami, Sabrina Naz, Sarah Khanlari, Anne McKenzie, Kingsley Agho, Andrew Page, Jane Ussher, Janette Perz, John Eastwood

**Affiliations:** 1Translational Health Research Institute, School of Medicine, Western Sydney University, Campbelltown Campus, Locked Bag 1797, Penrith, NSW 2571, Australia; 2General Practice Unit, Prescot Specialist Medical Centre, Welfare Quarters, Makurdi, Benue State 972261, Nigeria; 3Department of Community Paediatrics, Sydney Local Health District, Croydon Community Health Centre, 24 Liverpool Street, Croydon, NSW 2132, Australia; 4School of Medicine and Public Health, University of Newcastle, University Drive, Callaghan, NSW 2308, Australia; 5Primary & Community Health, Child and Family, South Western Sydney Local Health District, Narellan CHC, NSW 2567, Australia; 6Ingham Institute for Applied Medical Research, 1 Campbell Street, Liverpool, NSW 2170, Australia; 7School of Women’s and Children’s Health, The University of New South Wales, Kensington, Sydney, NSW 2052, Australia; 8Menzies Centre for Health Policy, Charles Perkins Centre, School of Public Health, Sydney University, Sydney, NSW 2006, Australia; 9Sydney Institute for Women, Children and their Families, Sydney Local Health District, Camperdown, NSW 2050, Australia

**Keywords:** Australia, culturally and linguistically diverse (CALD), depression, distress, perinatal

## Abstract

Perinatal distress and depression can have significant impacts on both the mother and baby. The present study investigated psychosocial and obstetric factors associated with perinatal distress and depressive symptoms among culturally and linguistically diverse (CALD) Australian women in Sydney, New South Wales. The study used retrospectively linked maternal and child health data from two Local Health Districts in Australia (*N* = 25,407). Perinatal distress was measured using the Edinburgh Postnatal Depression Scale (EPDS, scores of 10–12) and depressive symptoms, with EPDS scores of 13 or more. Multivariate multinomial logistic regression models were used to investigate the association between psychosocial and obstetric factors with perinatal distress and depressive symptoms. The prevalence of perinatal distress and depressive symptoms among CALD Australian women was 10.1% for antenatal distress; 7.3% for antenatal depressive symptoms; 6.2% for postnatal distress and 3.7% for postnatal depressive symptoms. Antenatal distress and depressive symptoms were associated with a lack of partner support, intimate partner violence, maternal history of childhood abuse and being known to child protection services. Antenatal distress and depressive symptoms were strongly associated with postnatal distress and depressive symptoms. Higher socioeconomic status had a protective effect on antenatal and postnatal depressive symptoms. Our study suggests that current perinatal mental health screening and referral for clinical assessment is essential, and also supports a re-examination of perinatal mental health policy to ensure access to culturally responsive mental health care that meets patients’ needs.

## 1. Introduction

Worldwide, depression is one of the leading causes of health loss, in terms of disability-adjusted life years. In 2017, depression was the second leading cause of disability-adjusted life years (after low back pain) among Australian women aged 15–49 years [1]. Perinatal depression (the occurrence of depression from conception to 12 months old postpartum) is a major mental health issue among women as it can have substantial health effects on both the mother and baby [1]. Evidence indicates that perinatal depression is associated with low birth weight, behavioural and cognitive problems, as well as delays in childhood language acquisition [2,3,4,5,6]. For the mother, the effects of perinatal depression may include non-adherence to antenatal care schedule, insomnia, poor appetite and weight gain, and poor social-emotional interaction with other family members. Low risk of suicidal ideation and suicide has also been documented in some women who reported perinatal depression [2,3,4,5].

Maternal distress plays a significant role in mother–infant interactions and can have adverse consequences for child growth and development [3,7]. Maternal distress can also amplify parenting stress, with subsequent impacts on the biopsychosocial profile of both the mother and her infant [8,9,10]. In Australia, despite frequent contact between women and healthcare practitioners in the perinatal period, the majority of women do not seek help for symptoms of perinatal distress [3,11,12,13]. Past studies have reported links between maternal distress and postnatal depression [14,15]. However, it is not clear whether maternal distress during pregnancy is related to postnatal distress and/or postnatal depression, particularly among culturally and linguistically diverse (CALD) Australian women; nor has there been studies that report on the relationship between psychosocial and obstetric factors with maternal distress during pregnancy among this subgroup of women.

Many Australian jurisdictions use the term ‘CALD’ to describe migrant and refugee communities with diverse ethnic background, traditions, food, nationality, language, dress, societal structures, art and religious characteristics [16,17]. The term is often used to characterise a subpopulation who are often socioeconomically disadvantaged for the purpose of focused interventions and research [18,19]. For example, evidence has shown that women from CALD backgrounds were more likely to report intimate partner violence [20] and depressive symptoms [21], and also less likely to use sexual and reproductive health services [22,23] compared to those from non-CALD backgrounds.

In most Australian primary health care settings, the Edinburgh Postnatal Depression Scale (EPDS) is routinely used to identify women who may be experiencing or are at risk of perinatal depression, and referral for health professional assessment and management. The use of the EPDS is also important in research relating to perinatal depression to inform targeted policies and practices [5,24]. Additionally, the utility of the EPDS to screen for perinatal distress and borderline personality disorder has also been described in recent Australian studies [25,26] and internationally [27].

In recent years, studies have examined the relationship between psychosocial and obstetric factors with perinatal depression among Australian woman based on the EPDS [14,21,28]. These efforts, although useful, did not specifically consider the association between these psychosocial and obstetric factors and perinatal depression among women from CALD backgrounds; nor did these studies capture emerging evidence on the impact of psychosocial and obstetric factors on maternal perinatal distress or assess the relationship between antenatal distress and postnatal distress or depression in this subgroup of women. Additionally, Australian mental health services usually do not adequately capture information on CALD subgroups [29,30], and thus, generalities of perinatal depression evidence can mask specifics, with the potential for policy-makers and practitioners to design interventions that may not meet patients’ needs.

To address these gaps in knowledge, the present study aimed to investigate the psychosocial and obstetric factors related to perinatal distress and depression among Australian CALD women in Sydney, New South Wales. In addition, the study will offer insight into the potential factors associated with perinatal distress and depression among CALD Australian women in order to inform culturally-respectful and focused interventions.

## 2. Methods 

### 2.1. Data Source

The overall methodology used in this study has been described elsewhere [21,31,32]. For this study, linked retrospective maternal and child health data of all live births in public health facilities among the population of CALD women in Sydney Local Health District (SLHD) and South Western Sydney Local Health District (SWSLHD) between 2014 and 2016 (*N* = 25,407) were used. These data were routinely collected as part of standard care provided to women during pregnancy and the postnatal period. Antenatal information (e.g., demographic characteristics, antenatal depressive symptoms using the EPDS and history of intimate partner violence, IPV) were collected by qualified midwives at the first prenatal care visit. Additionally, during the first prenatal visit, women were asked to identify whether they belong to CALD, non-CALD, or Aboriginal or Torres Strait Islander subpopulations. CALD population was defined based on the Australian Bureau of Statistics description [16]. Postnatal data (e.g., information on post-birth depressive symptoms based on the EPDS) were also collected during postnatal visits by qualified child and family health nurses. These maternal and child health data were stored in the Local Health District’s Information Management and Technology Division database. The data were obtained from the Information Management and Technology Division and linked using individual identifiers. 

### 2.2. Study Setting

The 2016 Australian Bureau of Statistics, Census of Population and Housing indicated that almost half of Australians (45%) were either born overseas or had one or both parents who were born overseas (19%) [33]. In Sydney, the SLHD and SWSLHD cover 52% of the metropolitan area, with an estimated population of 1.6 million people of different cultural backgrounds [34,35]. In the Sydney metropolitan area, more than half of the population spoke English at home (58.4%). Other most common languages spoken at home included Mandarin (4.7%), Arabic (4.0%) and Cantonese (2.9%) [36]. SLHD is located in the centre and inner west of Sydney, while SWSLHD is located in the south-western region of the city. A number of maternal and child health services are provided to all communities across both districts, including those with socioeconomically disadvantaged populations [34,35].

### 2.3. Study Factors

The variables were broadly categorised into psychosocial and obstetric factors, and selected based on past studies [14,21,28] and data availability. The psychosocial factors included maternal age (categorised as <20, 20–34, or ≥35 years); socioeconomic status (categorised as low, middle or high); partner support (categorised as yes or no); maternal history of childhood abuse (categorised as yes or no); history of psychological IPV (categorised as yes or no); history of physical IPV (categorised as yes or no); being known to Family and Community Services (FACS or child protection services, yes or no); previous child in out-of-home care (OOHC, categorised as yes or no); and major nationality groups based on the country of birth (categorised as Oceania, North-West Europe, Southern-Eastern Europe, North Africa and the Middle East, South-East Asia, North-East Asia, Southern and Central Asia, Americas or Sub-Saharan Africa).

Obstetric factors included the history of antenatal health problems (such as diabetes and/or hypertension, categorised as yes or no); alcohol use in pregnancy (categorised as yes or no), and type of delivery (categorised as normal vaginal, assisted vaginal or caesarean delivery). Information on the type of delivery was collected soon after birth, and data on other study factors were collected in the first postnatal visit.

Socioeconomic status was calculated using the Socio-Economic Index for Areas (SEIFA). SEIFA is an indicator created by the Australian Bureau of Statistics using principal component analysis and is an area-based scale in Australia according to socio-economic advantage and disadvantage. The variables used in the estimation of SEIFA cover a number of areas, including household income, employment, education, occupation and housing, as well as other indicators of advantage and disadvantage [37]. In the present study, deciles of socioeconomic status were categorised into high (top 10% of the population), middle (middle 80%) and low (bottom 10%) groups, similar to previously published studies [6,21]. In accordance with NSW Health policy [38], IPV information was collected from mothers based on the following question: (i) “within the last year have you been hit, slapped or hurt in other ways by your partner or ex-partner?”—physical IPV (ii) “are you frightened of your partner or ex-partner?”—psychological IPV. Maternal prenatal distress and depressive symptoms were also considered as potential factors associated with postnatal distress and depressive symptoms based on past studies [15,25]. 

### 2.4. Outcome Variables

The main outcome variables were antenatal distress and antenatal depressive symptoms, and postnatal distress and postnatal depressive symptoms, measured using the EPDS. The EPDS has been validated for use in the antenatal and postnatal periods to assess perinatal depressive symptoms in Australia and internationally [22,23,24]. In Australia, qualified midwives collect information on depressive symptoms at the first antenatal care visit using the EPDS. The total number of prenatal depressive symptoms are tallied to achieve a total score (out of 30). This score is entered into the Information Management and Technology Division database as a variable, which was categorised in the current study as ≤9, 10–12 or ≥13, with a score of 10–12 indicating distress and a score of 13 or more suggestive of maternal antenatal depressive symptoms [5,39,40]. Postnatal depressive symptoms were also collected during postnatal visits by child and family health nurses within the first 6-weeks of birth. Similar to antenatal depressive symptoms calculation, the overall number of postnatal depressive symptoms was calculated to obtain a score (out of 30), which was then categorised as ≤9, 10–12 or ≥13, with a score of 10–12 indicating distress and a score of 13 or more suggestive of postnatal depressive symptoms [10,39,40]. 

In the Local Health Districts, a woman whose responses indicate a higher EPDS score of ≥13 is referred to a clinician for formal assessment of depression and appropriate management, consistent with the NSW government guidelines on improving mental health outcomes for parents and infants [41]. However, women who score between 10 and 12 on the EPDS, signalling distress, are often underserved by current policy recommendations. Detailed information on the use of the EPDS in clinical practice and research in the Australian context has been published elsewhere [5,25,41]. In the current study, the EPDS cut-points used to indicate distress and depressive symptoms were based on previously published studies [6,15,25,28,42] and NSW government guidelines [41]. 

In the assessment of CALD women who could not communicate in English during the antenatal and postnatal periods, the English version of the EPDS was administered through qualified interpreters or via the use of non-English versions of the EPDS. The interpreters were certified by the National Accreditation Authority for Translators and Interpreters in Australia. The non-English versions of the EPDS were produced by the New South Wales Multicultural Health Communication Service. The EPDS has been translated and validated in a number of non-English speaking contexts [43], including studies of Iranian [44], Bangladeshi [45], Chinese [46], Serbian [47], and Greek women [48]. This population is part of the CALD community in the study cohort, where a high proportion of the women were from South East Asia (24.5%), North Africa and the Middle East (23.0%), and Southern and Central Asia (20.2%) [Appendix A].

### 2.5. Statistical Analysis

The initial analyses involved the calculation of frequencies of the study factors and outcomes (i.e., antenatal distress and depressive symptoms, and postnatal distress and depressive symptoms). Cross-tabulations of the study factors with the outcome variables were also conducted. This was followed by univariate regression models to investigate the relationship between each study factor and the outcome variables.

Multivariate multinomial logistic regression models were used to investigate the association between each study factor and the outcome variables, with adjustment for potential cofounders and year of the data (2014–2016). The data were combined to increase the statistical power of the study. In the models, adjustments for the birthing facility and gender of the baby, as well as relevant psychosocial and obstetric factors were conducted, consistent with previous studies [21,31]. Odds ratios (ORs) with corresponding 95% confidence intervals (CIs) were calculated as the measure of association between the study factors and perinatal distress and perinatal depressive symptoms.

We also examined the potential impact of missing data on the estimated measure of effect in sensitivity analyses that employed an imputed dataset. This was based on the original maternal and child health data which included complete data for perinatal distress and perinatal depressive symptoms. Multiple imputations by chained equations were employed, which assumes that data were missing at random [49]. This analytical approach also assumes that the known characteristics of study participants can be used to examine the characteristics of participants with missing data [50]. All study factors and outcome variables in the main analysis were included in the multiple imputation models. Revised odds ratios from the imputed data were estimated using the *mim* command, for comparison with the complete case analyses. Sensitivity analyses were conducted based on 25 multiple imputations [51], and all analyses were conducted in Stata (Stata Corp, version 15.0, College Station, TX, USA). 

### 2.6. Ethics

The Sydney Local Health District and South Western Sydney Local Health District Human Research Ethics Committees approved the collection of the data from the Information Management and Technology Division database and subsequent analysis. Approval numbers HREC: LNR/11/LPOOL/463; SSA: LNRSSA/11/LPOOL/464 and Project No: 11/276 LNR; Protocol No X12-0164 and LNR/12/RPAH/266. The data were accessed in accordance with the ethics protocol that only allowed unit record information to be released to investigators included in the ethics submission.

## 3. Results

### 3.1. Prevalence and Determinants of Antenatal Distress (Edinburgh Postnatal Depression Scale (EPDS) = 10–12)

The prevalence of antenatal distress was 10.1% among CALD women. The proportion of women who reported antenatal distress was higher in those who were known to FACS (20.6%), reported psychological IPV (18.8%) or lack of partner support (16.1%) compared to their counterparts (Table 1). CALD women who reported childhood abuse were more likely to report distress during the antenatal period compared to those who did not report abuse in the complete case analyses. Women who had a child in out-of-home care were more likely to report antenatal distress compared to their counterparts. Women from middle socioeconomic status category were less likely to experience antenatal distress compared to those from lower socioeconomic status category. Women who experienced psychological IPV were more likely to report distress during pregnancy compared to those who did not experience psychological IPV. The odds of antenatal distress were higher among CALD women who reported having an unsupportive partner, a history of antenatal medical problems or were known to child protection services (FACS) (Table 1).

### 3.2. Prevalence and Determinants of Antenatal Depressive Symptoms (EPDS ≥ 13) 

The prevalence of antenatal depressive symptoms was 7.3%; higher in women who reported having an unsupportive partner (33.3%), and those who reported psychological and physical IPV (26.6% and 30.9%, respectively) compared to their counterparts (Table 2). Women from middle socioeconomic status backgrounds were less likely to experience depressive symptoms during pregnancy compared to those from low socioeconomic status backgrounds in the complete case analyses. Women who were known to FACS and/or had a child in out-of-home care placement were more likely to experience antenatal depressive symptoms compared to those who were not known to FACS and/or did not report having a child in out-of-home care placement in the complete case analyses, respectively. The odds of maternal antenatal depressive symptoms were higher among women who reported a history of childhood abuse, psychological and physical IPV (Table 2).

### 3.3. Prevalence and Determinants of Postnatal Distress (EPDS = 10–12)

The proportion of CALD women who reported postnatal distress was 6.2%. The prevalence was higher among women who reported antenatal depressive symptoms (15.4%) compared to those who reported no antenatal depressive symptoms (Table 3). The study showed that women who experienced distress and depressive symptoms during pregnancy were more likely to report post-birth distress compared to those who reported no prenatal distress and depressive symptoms in the complete case analyses. Higher maternal socioeconomic status was associated with increased odds of postnatal distress (Table 3).

### 3.4. Prevalence and Determinants of Postnatal Depressive Symptoms (EPDS ≥ 13)

The prevalence of postnatal depressive symptoms was 3.7% in the study population. This proportion was higher among women who were known FACS compared to those who were not known (Table 4). The study indicated that women who experienced maternal distress and depressive symptoms during pregnancy were more likely to report post-birth depressive symptoms compared to those who reported no prenatal distress and depressive symptoms in the complete case analyses (Table 4). The odds of postnatal depressive symptoms were lower among CALD women from high socioeconomic status category but higher in those who reported a history of childhood abuse, being known to child protection services and a history of physical IPV. Women were more likely to experience postnatal depressive symptoms if they had assisted vaginal birth or caesarean section, or were older (≥35 years) in the complete case analyses (Table 4).

There were wide-ranging relationships between major nationality groups and perinatal distress and depressive symptoms (Table 1, Table 2, Table 3 and Table 4). The association between high socioeconomic status group was associated with antenatal distress in the imputed data compared to that in the complete case analyses. A history of having a child in out-of-home care setting was associated with antenatal depressive symptoms in the imputed data compared to that in the complete case analyses Similar findings were also observed in the relationship between some of the study factors and postnatal distress and depressive symptoms (Table 3 and Table 4). Additionally, the corresponding 95% CIs of the ORs for some of the study factors based on the imputed data are relatively narrower than those obtained based on the original data set, potentially suggesting the effect of missing data or small sample size (Table 1, Table 2, Table 3 and Table 4).

## 4. Discussion

The study investigated the prevalence and psychosocial and obstetric determinants of perinatal distress and depressive symptoms among CALD Australian women who resided in Sydney, Australia between 2014 and 2016. The prevalence of antenatal distress was 10.1%, while that for depressive symptoms during pregnancy was 7.3%. The proportion of women who reported post-birth distress was 6.2%, while those who reported postnatal depressive symptoms was 3.7%. The present study also indicated that the psychosocial and obstetric factors associated with perinatal distress and depression varied, which potentially reflect the interaction between many individual-level factors, including socioeconomic status, culture and acculturation (the process of adjusting to a new culture) [52].

Consistent with past reports [53,54], the study highlighted a range of complex factors that make it challenging for CALD women to achieve better health outcomes. The factors associated with maternal distress and depressive symptoms during pregnancy included a maternal history of childhood abuse, having a child in out-of-home care setting, being known to child protection services, IPV and antenatal health problems. For postnatal distress, associated factors included maternal distress and depressive symptoms during pregnancy. Distress and depressive symptoms during pregnancy, maternal history of childhood abuse, being known to child protection services, physical IPV, and assisted vaginal birth or caesarean section were factors associated with postnatal depressive symptoms. Many of these individual-level factors have a complex interaction with each other. For example, reports from Australia have shown that women who reported IPV and/or those who resided in public housing were more likely to experience homelessness [55,56]. Notably, most of these factors are related to the key concept of “community disadvantage”, which develops from the interaction between individual-level characteristics (e.g., limited education and drug and alcohol use) and the impacts of social and environmental settings (such as weak social networks and a relative lack of opportunity) [53]. Past studies conducted in Australia have documented associations between these multifaceted psychosocial factors and psychiatric illness [57,58]. In New South Wales, Australia, recent appreciation and understanding of the interplay between these factors have increased the potential for policymakers and public health experts to design effective and integrated perinatal mental health interventions [59,60]. These improvements also meant that practitioners can provide comprehensive, holistic and patient-centred care to CALD women who may be experiencing or are at risk for perinatal distress or depressive symptoms.

Worldwide, interventions that target the socioeconomic status of women are one of the cornerstones for improving maternal and child health outcomes [61]. Studies conducted in Australia have shown that there is a strong association between low socioeconomic status and the onset of psychological distress and mental disorders [62,63], and other adverse health outcomes [64,65]. The present study showed that middle socioeconomic status category was associated with a reduced risk of maternal distress and depressive symptoms during pregnancy; this association was also significant for high socioeconomic status women in the imputed dataset. Similarly, high socioeconomic status had a protective effect on postnatal depressive symptoms. In contrast, high socioeconomic status category was associated with postnatal distress. In Australia, there is limited epidemiological data on perinatal mental health among CALD women that may help contextualise and possibly explain why high socioeconomic status women were more likely to experience postnatal distress compared to those from low socioeconomic status category [29,66]. Nevertheless, there are other key factors that may interact with socioeconomic status (including anxiety [67], stress [68], acculturation [52], racism and discrimination [69,70]) to affect the study outcomes. Whether or not these factors and their interplay with each other are relevant to the perinatal mental health of CALD Australian women from the high-income category is unclear. Additionally, socioeconomic status also plays a key role within the context of “community disadvantage”, to contribute additional burden to vulnerable populations. Future epidemiological studies that examine the impact of socioeconomic status inequalities on perinatal mental health of CALD Australian women may be needed to guide policy decision-makers and public health practitioners design effective and focused interventions. 

The present study indicated that antenatal health issues were associated with prenatal distress and depressive symptoms. Also, assisted vaginal delivery and antenatal health issues were associated with postnatal depressive symptoms, consistent with a previous study [21]. These factors have the potential to increase the health burden of women who may be struggling to cope with the stressful event of pregnancy and the arrival of the newborn. The study also showed different associations between major nationality groups and perinatal distress and depressive symptoms. For example, CALD women from European and Asian regions were less likely to experience antenatal depressive symptoms but those from North Africa and the Middle East were more likely to report prenatal depressive symptoms, compared to their counterparts from Oceania. Future studies that examine the impact of socioeconomic determinants on perinatal mental health among sub-groups within CALD communities may be needed as they would provide more opportunities for culturally responsive care and support.

### Policy Implications of the Study Findings

Our study has policy and practice implications for public health interventions and clinical practice as it underpins efforts to re-assess the New South Wales perinatal mental health implementation model—the SAFE START strategic policy [71]—and potentially Australian-wide perinatal mental health policy [29,72]. Detailed information on the broader rationale for calls to stakeholders to re-examine the current perinatal mental health policy has been published elsewhere [25]. Here, we highlight a current CALD mental health strategy and describe a number of available low cost, population-based interventions, and possibly culturally appropriate that may be initiated and expanded to improve the perinatal mental health of CALD Australian women. 

In 2018, the Australian Minister for Health announced a funding package for an alliance between Mental Health Australia, the Federation of Ethnic Communities’ Councils of Australia and the National Ethnic Disability Alliance to deliver a new National Multicultural Mental Health Project [72]. The collaboration has further developed the *Framework for Mental Health in Multicultural Australia: Towards culturally inclusive service delivery*, which is regarded as the cornerstone to coordinate mental health services for CALD consumers, carers and their families. Other essential components of this initiative include organisational and individual appraisal to achieve cultural responsiveness for CALD communities and support for collaboration between CALD mental health consumers, carers, community and the mental health sector [29]. Our study provides context-specific evidence on determining factors for perinatal distress and depressive symptoms among CALD women to support the continued engagement of perinatal mental health stakeholders, and underscores the rationale for renewed calls to re-examine the state-wide perinatal mental health policy [25]. In Australia and internationally, there are evidence-based treatment approaches to care for women experiencing depression [3,4]. These strategies include psychosocial support, psychological therapies (such as cognitive behavioural therapy, CBT) and pharmacotherapy, but the choice of treatment often depends on the severity of the woman’s symptoms, the presence of comorbidities, patient’s preferences and the options available [3,4]. Consequently, treatment outcomes may also differ among women depending on the presence or absence of those factors. Nonetheless, studies that, in particular, evaluate the effect of a single or complex intervention for women who signal distress in the perinatal period are limited in Australia. Khanlari et al [25] recently highlighted some low-cost, population-based initiatives that may be considered for Australian women (including those from CALD backgrounds) who signal perinatal distress on the EPDS.

The present study indicated that a lack of partner support was associated with antenatal distress and depressive symptoms, consistent with past research [21,73]. Couple-based approaches may play an important role in the prevention of distress and depression in the perinatal period as it involves the disclosure of concerns and feelings between partners. Couple-based interventions are based on the social cognitive model of stress and coping which suggests that stressful events (such as pregnancy) are a risk to the individual’s current plans and relationships [74]. Emerging evidence on the efficacy of couple-based interventions has been documented in patients living with cancer [75], psychological distress and mental illness such as depression [76,77]. Providing CALD couples with appropriate and adaptive communication skills that help them to make sense of the challenges of pregnancy, prepare for the arrival of the newborn and negotiate to change household responsibilities, as well as a coordinated coping response may be needed to improve perinatal mental health outcomes. 

Furthermore, there is emerging evidence from Australia [78] and international research [79,80] that suggests that internet-delivered psychoeducation (e.g., internet-delivered CBT, iCBT) may be an effective treatment strategy for perinatal distress and depression. However, whether the iCBT treatment option is fully applicable to CALD Australian women remains unclear, as the Australian study lacked active controls and largely involved partnered, well-educated women, with implications for the generalizability of the study results [78]. Well-designed trials that investigate the impact of iCBT on perinatal distress and depression and consider larger and more diverse study participants are needed. Additional information on future research directions on emerging interventions for perinatal distress and depression has been highlighted elsewhere [79].

The study has a number of limitations. First, the EPDS is a screening tool, and may not identify all mothers with depressive symptoms. Women with high scores may not have clinical depression, while those with low scores may underestimate or overestimate their symptoms. Second, the utility of the EPDS as a screening tool for depression may be challenging, as the assessment of postnatal distress or depressive symptoms may be due to the extra post-birth responsibilities associated with the arrival of a newborn. Third, in the present study, the EPDS was based on self-report in the seven days prior to contact with the health practitioner in the perinatal period, potentially leading to recall and/or measurement bias. This may have resulted in an overestimation or underestimation of the association between the study factors and the outcomes. Fourth, we were unable to differentiate women who have a prior history of mental illness (such as anxiety and/or depression) in the perinatal period. This analysis would have provided detailed information on the role of the study factors on the outcomes among CALD Australian women. Fifth, the study was unable to assess or adjust for other potential determining factors (e.g., prematurity, level of support received during pregnancy and postnatally or multi-parity) as this may also affect the observed estimates. Sixth, the small sample size available for some of the study factors may account for the large effect sizes and the wide CIs. Seventh, the non-use of maternal and child health data from private healthcare and other Local Health Districts in Sydney is a limitation in the present study, as it affects the external validity. Finally, the unavailability of longitudinal data from conception to 12-months post-birth and the use of secondary data were additional limitations in this study.

Despite these limitations, the study has strengths. First, the EPDS is an extremely useful tool for researchers and clinicians to evaluate the impact of preventive programs and identify women who may be experiencing or are at risk for perinatal distress or depressive symptoms [81,82]. Second, the EPDS is designed for use in the perinatal period as it does not assess somatic depressive symptoms such as weight change and sleep difficulties that often accompany uncomplicated pregnancies [83]. The study also provides specific epidemiological data on perinatal distress and depressive symptoms among CALD Australian women in Sydney, New South Wales, to support calls for the re-assessment of the perinatal mental health policy [25].

## 5. Conclusions

Our study showed varied proportions of perinatal distress and depressive symptoms among CALD Australian women who resided in Sydney, New South Wales. These proportions were higher among subgroups of women who reported psychological and physical IPV, antenatal depressive symptoms and those who were known to FACS. A lack of partner support, psychological and physical IPV, maternal history of childhood abuse and being known to FACS increased the risk of antenatal distress and antenatal depressive symptoms. Antenatal distress and antenatal depressive symptoms were the common factors associated with postnatal distress and postnatal depressive symptoms. The study suggests that screening for distress and depressive symptoms in the perinatal period and appropriate referral for clinical assessment may be useful to promptly identify and support CALD Australian women who may be experiencing or at-risk of perinatal distress or depressive symptoms.

## Figures and Tables

**Table 1 ijerph-16-02945-t001:** Associations between the study factors and antenatal distress (EPDS = 10–12) in South Western Sydney and Sydney Local Health Districts, 2014–2016.

				Complete Cases				Multiple Imputation *			
	Participants	Cases	%	Unadjusted OR (95%CI)	P Value	Adjusted OR (95% CI) ^#^	P Value	Unadjusted OR (95% CI)	P Value	Adjusted OR (95% CI) ^#^	P Value
Antenatal distress	20,560	2078	10.1								
Psychosocial factors											
Maternal age group											
20–34 years	19,333	1955	10.1	1.0		1.0		1.0		1.0	
≥35 years	1134	113	10.0	1.0 (0.8–1.2)	0.941	1.1 (0.8–1.4)	0.460	1.0 (0.8–1.2)	0.941	1.1 (0.8–1.4)	0.471
<20 years	93	10	10.8	1.2 (0.6–2.3)	0.617	1.6 (0.4–5.8)	0.439	1.2 (0.6–2.3)	0.617	1.7 (0.4–5.8)	0.430
Socioeconomic status category											
Low	10,385	1143	11.0	1.0		1.0		1.0		1.0	
Middle	7606	666	8.7	0.7 (0.7–0.8)	<0.001	0.8 (0.7–0.9)	0.006	0.7 (0.7–0.8)	<0.001	0.8 (0.7–0.9)	0.001
High	1336	103	7.7	0.6 (0.5–0.7)	<0.001	0.7 (0.5–1.0)	0.096	0.6 (0.5–0.8)	<0.001	0.7 (0.5–0.9)	0.042
Supportive partner											
Yes	19,491	1929	9.9	1.0		1.0		1.0		1.0	
No	559	90	16.1	2.6 (2.1–3.4)	<0.001	3.0 (2.1–4.3)	<0.001	2.6 (2.1–3.3)	<0.001	3.2 (2.3–4.5)	<0.001
Maternal history of childhood abuse											
No	19,102	1873	9.8	1.0		1.0		1.0		1.0	
Yes	708	109	15.4	2.0 (1.6–2.4)	<0.001	1.9 (1.4–2.6)	<0.001	2.0 (1.6–2.5)	<0.001	1.9 (1.4-2.6)	<0.001
Psychological IPV											
No	19,759	1953	9.9	1.0		1.0		1.0		1.0	
Yes	308	58	18.8	2.9 (2.1–3.9)	<0.001	2.8 (1.7–4.5)	<0.001	2.9 (2.1–3.9)	<0.001	2.6 (1.6-4.1)	<0.001
Physical IPV											
No	19,757	1965	10.0	1.0		1.0		1.0		1.0	
Yes	259	39	15.1	2.3 (1.6–3.3)	<0.001	0.9 (0.5–1.7)	0.950	2.3 (1.6–3.3)	<0.001	1.1 (0.6–1.8)	0.830
Mother was known to FACS											
No	17,188	1702	9.9	1.0		1.0		1.0		1.0	
Yes	126	26	20.6	3.1 (2.0–4.9)	<0.001	2.0 (1.1–3.8)	0.018	3.1 (2.0–5.0)	<0.001	2.0 (1.1–3.6)	0.012
Previous child in OOHC											
No	12,988	1218	9.3	1.0		1.0		1.0		1.0	
Yes	446	71	15.9	1.9 (1.5–2.5)	<0.001	1.6 (1.1–2.2)	0.003	2.0 (1.5–2.6)	<0.001	1.6 (1.2–2.2)	0.002
Major nationality group											
Oceania	1186	133	11.2	1.0		1.0					
North-West Europe	312	18	5.8	0.4 (0.2–0.7)	0.002	0.3 (0.1–0.7)	0.008	0.4 (0.4–0.5)	<0.001	0.3 (0.2–0.4)	<0.001
Southern-Eastern Europe	1066	55	5.2	0.3 (0.2–0.5)	<0.001	0.4 (0.3–0.7)	0.001	0.4 (0.4–0.4)	<0.001	0.4 (0.4–0.5)	<0.001
North Africa and the Middle East	4801	545	11.4	1.0 (0.8–1.2)	0.684	1.1 (0.8–1.5)	0.712	1.0 (1.0–1.1)	0.038	1.0 (1.0–1.1)	0.049
South-East Asia	5077	463	9.1	0.7 (0.6–0.9)	0.007	0.6 (0.4–0.8)	0.002	0.7 (0.7–0.8)	<0.001	0.6 (0.6–0.7)	<0.001
North-East Asia	2692	212	7.9	0.6 (0.5–0.7)	<0.001	0.6 (0.4–0.8)	0.005	0.6 (0.6–0.7)	<0.001	0.6 (0.5–0.6)	<0.001
Southern and Central Asia	4234	540	12.7	1.1 (0.9–1.4)	0.187	1.2 (0.9–1.6)	0.205	1.1 (1.1–1.2)	<0.001	1.1 (1.1–1.2)	<0.001
Americas	487	36	7.4	0.6 (0.4–0.8)	0.010	0.6 (0.3–1.1)	0.182	0.6 (0.5–0.6)	0.012	0.6 (0.5–0.7)	<0.001
Sub-Saharan Africa	705	76	10.7	0.9 (0.6–1.2)	0.621	0.9 (0.6–1.4)	0.794	0.9 (0.8–0.9)	<0.001	0.9 (0.9–1.0)	0.560
Obstetric factors											
Alcohol use in pregnancy											
No	20,049	2011	10.0	1.0		1.0		1.0		1.0	
Yes	163	25	15.3	1.6 (1.0–2.4)	0.028	1.6 (1.1–2.3)	0.011	1.6 (1.0–2.4)	0.031	1.6 (0.8–2.9)	0.160
Antenatal health problems											
No	16,364	1589	9.7	1.0		1.0		1.0		1.0	
Yes	3501	410	11.7	1.3 (1.1–1.4)	<0.001	1.2 (1.1–1.4)	0.002	1.3 (1.1–1.4)	<0.001	1.3 (1.1–1.6)	0.001

EPDS: Edinburgh Postnatal Depression Scale; OOHC: out of home care; FACS: Family and Community Services; IPV: intimate partner violence; ^#^ Multivariate models adjusted for the birthing facility and gender of the baby, as well as relevant psychosocial and obstetric factors. * Sensitivity analyses following multiple imputations for missing values; OR (95% CI): Odds ratio with 95% confidence interval.

**Table 2 ijerph-16-02945-t002:** Associations between the study factors and antenatal depressive symptoms (EPDS ≥ 13) in South Western Sydney and Sydney Local Health Districts, 2014–2016.

				Complete Cases				Multiple Imputation *			
	Participants	Cases	%	Unadjusted OR (95% CI)	P Value	Adjusted OR (95% CI) ^#^	P Value	Unadjusted OR (95% CI)	P Value	Adjusted OR (95% CI) ^#^	P Value
Antenatal depressive symptoms	20,560	1510	7.3								
Psychosocial factors											
Maternal age group											
20–34 years	19,333	1392	7.2	1.0		1.0		1.0			
≥35 years	1134	104	9.2	1.3 (1.1–1.6)	0.014	1.2 (0.9–1.6)	0.095	1.3 (1.1–1.6)	0.014	1.2 (0.9–1.6)	0.159
<20 years	93	14	15.1	2.3 (1.3–4.1)	0.004	1.2 (0.2–1.2)	0.805	2.3 (1.3–4.1)	0.004	2.1 (0.5–7.6)	0.246
Socioeconomic status category											
Low	10,385	915	8.8	1.0		1.0		1.0		1.0	
Middle	7606	456	6.0	0.6 (0.6–0.7)	<0.001	0.7 (0.5–0.8)	<0.001	0.6 (0.6–0.7)	<0.001	0.7 (0.6–0.8)	<0.001
High	1336	59	4.4	0.4 (0.3–0.5)	<0.001	0.5 (0.3–0.7)	0.001	0.4 (0.3–0.6)	<0.001	0.5 (0.3–0.7)	<0.001
Supportive partner											
Yes	19,491	1263	6.5	1.0	<0.001	1.0		1.0		1.0	
No	559	186	33.3	8.5 (6.9–10.2)	<0.001	6.7 (4.9–9.1)	<0.001	8.0 (6.6–9.7)	<0.001	6.5 (4.9–8.7)	<0.001
Maternal history of childhood abuse											
No	19,102	1288	6.7	1.0		1.0		1.0		1.0	
Yes	708	137	19.4	3.6 (3.0–4.4)	<0.001	3.2 (2.4–4.3)	<0.001	3.6 (3.0–3–4.4)	<0.001	3.1 (2.3–4.2)	<0.001
Psychological IPV											
No	19,759	1382	7.0	1.0		1.0		1.0		1.0	
Yes	308	82	26.6	5.8 (4.4–7.5)	<0.001	1.8 (1.0–3.1)	0.025	5.8 (4.4–7.5)	<0.001	1.9 (1.1–3.1)	0.010
Physical IPV											
No	19,757	1370	6.9	1.0		1.0		1.0		1.0	
Yes	259	80	30.9	4.7 (3.9–5.7)	<0.001	2.6 (1.5–4.4)	<0.001	6.8 (5.1–9.0)	<0.001	2.6 (1.5–4.3)	<0.001
Mother was known to FACS											
No	17,188	1201	7.0	1.0		1.0		1.0		1.0	
Yes	126	31	24.6	5.3 (3.4–8.1)	<0.001	2.3 (1.2–4.3)	0.006	5.3 (3.4–8.1)	<0.001	1.9 (1.0–3.4)	0.033
Previous child in OOHC											
No	12,988	903	7.0	1.0		1.0		1.0		1.0	
Yes	446	57	12.8	2.1 (1.6–2.8)	<0.001	1.4 (0.9–2.0)	0.378	2.1 (1.6–2.8)	<0.001	1.4 (1.0–2.0)	0.043
Major nationality group											
Oceania	1186	112	9.4	1.0		1.0		1.0		1.0	
North-West Europe	312	5	1.6	0.1 (0.1–0.3)	<0.001	0.1 (0.04–0.7)	0.017	0.1 (0.1–0.2)	<0.001	0.1 (0.1–0.2)	<0.001
Southern-Eastern Europe	1066	29	2.7	0.2 (0.1–0.3)	<0.001	0.2 (0.1–0.5)	<0.001	0.2 (0.2–0.3)	<0.001	0.2 (0.2–0.3)	<0.001
North Africa and the Middle East	4801	559	11.6	1.2 (1.0–1.5)	0.029	1.7 (1.2–2.3)	0.001	1.2 (1.2–1.3)	<0.001	1.6 (1.5–1.7)	<0.001
South-East Asia	5077	263	5.2	0.5 (0.4–0.6)	<0.001	0.5 (0.4–0.8)	0.002	0.5 (0.4–0.5)	<0.001	0.5 (0.5–0.6)	<0.001
North-East Asia	2692	103	3.8	0.3 (0.2–0.4)	<0.001	0.4 (0.3–0.7)	0.001	0.3 (0.3–0.4)	<0.001	0.4 (0.4–0.5)	<0.001
Southern and Central Asia	4234	362	8.6	0.9 (0.7–1.1)	0.425	1.1 (0.8–1.6)	0.373	0.9 (0.8–0.9)	<0.001	1.1 (1.0–1.2)	<0.001
Americas	487	28	5.8	0.5 (0.3–0.8)	0.007	0.7 (0.4–1.4)	0.438	0.5 (0.5–0.6)	<0.001	0.7 (0.6–0.8)	<0.001
Sub-Saharan Africa	705	49	7.0	0.7 (0.4–1.0)	0.056	0.5 (0.3–0.9)	0.044	0.7 (0.6–0.7)	<0.001	0.5 (0.4–0.6)	<0.001
Obstetric factors											
Alcohol use in pregnancy											
No	20,049	1471	7.3	1.0		1.0		1.0		1.0	
Yes	163	11	6.8	0.9 (0.5–1.8)	0.937	0.6 (0.2–1.7)	0.413	0.9 (0.5–1.8)	0.962	0.9 (0.3–2.1)	0.853
Antenatal health problems											
No	16,364	1129	6.9	1.0		1.0		1.0		1.0	
Yes	3501	309	8.8	1.3 (1.2–1.5)	<0.001	1.3 (1.0–1.6)	0.011	1.3 (1.2–1.5)	<0.001	1.3 (1.0–1.6)	0.010

EPDS: Edinburgh Postnatal Depression Scale; OOHC: out of home care; FACS: Family and Community Services; IPV: intimate partner violence; ^#^ Multivariate models adjusted for the birthing facility and gender of the baby, as well as relevant psychosocial and obstetric factors. * Sensitivity analyses following multiple imputations for missing values; OR (95% CI): Odds ratio with 95% confidence interval.

**Table 3 ijerph-16-02945-t003:** Associations between the study factors and postnatal distress (EPDS = 10–12) in South Western Sydney and Sydney Local Health Districts, 2014–2016.

				Complete Cases				Multiple Imputation *			
	Participants	Cases	%	Unadjusted OR (95% CI)	P Value	Adjusted OR (95% CI) ^#^	P Value	Unadjusted OR (95% CI)	P Value	Adjusted OR (95% CI) ^#^	P Value
Postnatal distress	19,342	1194	6.2								
Psychosocial factors											
Antenatal EPDS											
EPDS ≤ 9	13,075	559	4.3	1.0		1.0		1.0		1.0	
EPDS 10–12	1633	213	13.2	3.6 (3.0–4.2)	<0.001	4.0 (3.1–5.2)	<0.001	2.8 (2.4–3.3)	<0.001	3.0 (2.3–3.8)	<0.001
EPDS ≥ 13	1189	183	15.4	4.9 (4.1–5.9)	<0.001	5.5 (4.1–7.3)	<0.001	3.7 (3.1–4.4)	<0.001	3.9 (3.0–5.1)	<0.001
Maternal age group											
20–34 years	18,147	1114	6.1	1.0		1.0		1.0		1.0	
≥35 years	1106	74	6.7	1.1 (0.8–1.4)	0.366	1.1 (0.7–1.6)	0.460	1.0 (0.8–1.3)	0.493	1.1 (0.7–1.5)	0.577
<20 years	89	6	6.7	1.1 (0.4–2.5)	0.795	0.1 (0.1–9.4)	0.885	1.1 (0.5–2.3)	0.770	0.9 (0.1–6.7)	0.922
Socioeconomic status category											
Low	9629	584	6.1	1.0		1.0		1.0		1.0	
Middle	7389	437	5.9	0.9 (0.8–1.0)	0.516	1.1 (0.9–1.3)	0.276	0.9 (0.8–1.0)	0.584	1.0 (0.8–1.3)	0.416
High	1345	103	7.7	1.2 (0.9–1.5)	0.042	1.6 (1.1–2.3)	0.009	1.1 (0.9–1.4)	0.169	1.4 (1.0–1.9)	0.050
Supportive partner											
Yes	16,275	967	5.9	1.0		1.0		1.0		1.0	
No	451	49	10.9	2.1 (1.5–2.9)	<0.001	1.2 (0.7–2.1)	0.379	2.1 (1.5–2.9)	<0.001	1.3 (0.8–2.2)	0.224
Maternal history of childhood abuse											
No	15,846	931	5.9	1.0		1.0		1.0		1.0	
Yes	607	61	10.1	1.9 (1.4–2.5)	<0.001	1.2 (0.7–1.9)	0.328	1.7 (1.3–2.2)	<0.001	1.1 (0.7–1.8)	0.532
Psychological IPV											
No	16,420	982	5.9	1.0		1.0		1.0		1.0	
Yes	264	26	9.9	1.8 (1.2–2.8)	0.003	1.0 (0.4–2.2)	0.997	1.7 (1.2–2.6)	0.004	1.0 (0.5–2.2)	0.835
Physical IPV											
No	16,418	991	6.0	1.0		1.0		1.0		1.0	
Yes	220	15	6.8	1.2 (0.7–2.2)	0.334	0.8 (0.3–2.2)	0.734	1.3 (0.8–2.1)	0.262	0.7 (0.2–1.8)	0.481
Mother was known to FACS											
No	14,357	845	5.9	1.0		1.0		1.0		1.0	
Yes	103	11	10.7	2.3 (1.2–4.5)	0.008	1.8 (0.8–4.4)	0.144	1.9 (1.0–3.6)	0.038	1.5 (0.6–3.4)	0.321
Previous child in OOHC											
No	10,639	590	5.6	1.0		1.0		1.0		1.0	
Yes	345	19	5.5	1.0 (0.6–1.6)	0.966	0.6 (0.3–1.4)	0.250	0.9 (0.6–1.5)	0.951	0.7 (0.4–1.3)	0.328
Major nationality group											
Oceania	1143	59	5.2	1.0		1.0		1.0		1.0	
North–West Europe	284	15	5.3	1.0 (0.5–1.8)	0.929	1.4 (1.0–1.5)	0.525	0.9 (0.9–1.1)	0.983	1.1 (0.9–1.3)	0.169
Southern–Eastern Europe	1030	49	4.8	0.8 (0.6–1.3)	0.585	1.3 (1.0–1.5)	0.453	0.9 (0.8–0.9)	0.036	1.1 (1.0–1.2)	0.027
North Africa and the Middle East	4218	271	6.4	1.2 (0.9–1.7)	0.083	1.9 (1.0–1.5)	0.014	1.2 (1.1–1.2)	<0.001	1.4 (1.3–1.6)	<0.001
South–East Asia	4940	282	5.7	1.1 (0.8–1.4)	0.478	1.6 (1.0–1.5)	0.074	1.0 (1.0–1.1)	0.006	1.2 (1.1–1.4)	<0.001
North–East Asia	2540	152	6.0	1.1 (0.8–1.5)	0.334	1.8 (1.0–1.5)	0.035	1.1 (1.0–1.1)	<0.001	1.4 (1.3–1.5)	<0.001
Southern and Central Asia	4011	298	7.4	1.4 (1.1–1.9)	0.007	2.0 (1.0–1.5)	0.008	1.3 (1.2–1.4)	<0.001	1.6 (1.5–1.8)	<0.001
Americas	501	32	6.4	1.2 (0.7–1.9)	0.365	1.2 (1.0–1.5)	0.630	1.1 (1.0–1.2)	<0.001	1.1 (0.9–1.2)	0.096
Sub–Saharan Africa	675	36	5.3	1.0 (0.6–1.6)	0.835	2.6 (1.0–1.5)	0.004	1.0 (0.9–1.1)	0.238	1.8 (1.6–2.1)	<0.001
Obstetric factors											
Alcohol use in pregnancy											
No	17,001	1039	6.1	1.0		1.0		1.0		1.0	
Yes	143	13	9.1	1.5 (0.8–2.6)	0.158	1.4 (0.5–3.8)	0.413	1.5 (0.8–2.6)	0.156	1.7 (0.8–3.9)	0.148
Antenatal health problems											
No	15,755	960	6.1	1.0		1.0		1.0		1.0	
Yes	3041	206	6.8	1.1 (0.9–1.3)	0.114	1.2 (0.9–1.6)	0.084	1.0 (0.9–1.2)	0.234	1.1 (1.1–1.2)	<0.001
Type of delivery											
Normal vaginal	11,336	602	5.3	1.0		1.0		1.0		1.0	
Assisted vaginal	2323	176	7.6	1.4 (1.2–1.7)	<0.001	1.2 (0.8–4.4)	0.175	1.3 (1.1–1.6)	0.001	1.3 (0.9–1.8)	0.092
Caesarean section	5658	415	7.3	1.4 (1.2–1.6)	<0.001	1.1 (0.9–1.4)	0.262	1.3 (1.1–1.5)	<0.001	1.1 (0.8–1.3)	0.372

EPDS: Edinburgh Postnatal Depression Scale; OOHC: out of home care; FACS: Family and Community Services; IPV: intimate partner violence; ^#^ Multivariate models adjusted for the birthing facility and gender of the baby, as well as relevant psychosocial and obstetric factors. * Sensitivity analyses following multiple imputations for missing values; OR (95% CI): Odds ratio with 95% confidence interval.

**Table 4 ijerph-16-02945-t004:** Associations between the study factors and postnatal depressive symptoms (EPDS ≥ 13) in South Western Sydney and Sydney Local Health Districts, 2014–2016.

				Complete Cases				Multiple Imputation *			
	Participants	Cases	%	Unadjusted OR (95% CI)	P Value	Adjusted OR (95%CI) ^#^	P Value	Unadjusted OR (95% CI)	P Value	Adjusted OR (95% CI) ^#^	P Value
Postnatal depressive symptoms	19,342	723	3.7								
Psychosocial factors											
Antenatal EPDS											
EPDS ≤ 9	13,075	268	2.1	1.0		1.0		1.0		1.0	
EPDS 10–12	1633	112	6.9	3.9 (3.1–4.9)	<0.001	4.5 (3.2–6.3)	<0.001	2.9 (2.3–3.6)	<0.001	3.1 (2.2–4.4)	<0.001
EPDS ≥ 13	1189	201	16.9	11.4 (9.3–13.8)	<0.001	11.4 (9.3–16.1)	<0.001	7.6 (6.3–9.2)	<0.001	7.3 (5.4–9.9)	<0.001
Maternal age group											
20–34 years	18,147	659	3.6	1.0		1.0		1.0		1.0	
≥35 years	1106	60	5.4	1.5 (1.1–2.0)	0.002	1.5 (1.0–2.4)	0.026	1.3 (1.0–1.8)	0.019	1.3 (0.9–2.0)	0.113
<20 years	89	4	4.5	1.2 (0.4–3.4)	0.655	1.4 (0.1–12.3)	0.748	1.2 (0.45–3.4)	0.659	1.0 (0.1–9.2)	0.953
Socioeconomic status category											
Low	9629	432	4.5	1.0		1.0		1.0		1.0	
Middle	7389	228	3.1	0.6 (0.5–0.7)	<0.001	0.8 (0.86–1.0)	0.103	0.7 (0.6–0.8)	<0.001	0.8 (0.6–1.0)	0.105
High	1345	30	2.2	0.4 (0.3–0.7)	0.008	0.4 (0.2–0.9)	0.036	0.5 (0.4–0.8)	0.002	0.5 (0.2–1.0)	0.078
Supportive partner											
Yes	16,275	555	3.4	1.0		1.0		1.0		1.0	
No	451	53	11.8	4.0 (2.9–5.4)	<0.001	1.2 (0.7–2.1)	0.399	3.9 (2.9–5.3)	<0.001	1.7 (1.1–2.9)	0.018
Maternal history of childhood abuse											
No	15,846	540	3.4	1.0		1.0		1.0		1.0	
Yes	607	52	8.6	2.8 (2.0–3.7)	<0.001	2.0 (1.2–3.1)	0.003	2.4 (1.7–3.3)	<0.001	1.6 (1.0–2.5)	0.044
Psychological IPV											
No	16,420	581	3.5	1.0		1.0		1.0		1.0	
Yes	264	29	11.0	3.5 (2.3–5.2)	<0.001	1.0 (1.0–3.2)	0.979	3.0 (2.0–4.5)	<0.001	0.9 (0.4–1.9)	0.923
Physical IPV											
No	16,418	573	3.5	1.0		1.0		1.0		1.0	
Yes	220	32	14.6	4.7 (3.2–7.0)	<0.001	2.5 (1.2–5.4)	0.015	3.9 (2.6–5.7)	<0.001	2.0 (1.0–4.1)	0.037
Mother was known to FACS											
No	14,357	488	3.4	1.0		1.0		1.0		1.0	
Yes	103	21	20.4	7.8 (4.8–12.9)	<0.001	4.0 (1.9–8.4)	<0.001	5.5 (3.4–9.0)	<0.001	2.9 (1.4–5.8)	0.002
Previous child in OOHC											
No	10,639	355	3.3	1.0		1.0		1.0		1.0	
Yes	345	17	4.9	1.5 (0.9–2.4)	0.110	0.7 (0.3–1.3)	0.315	1.3 (0.8–2.1)	0.182	0.8 (0.4–1.4)	0.503
Major nationality group											
Oceania	1143	38	3.3	1.0		1.0		1.0		1.0	
North–West Europe	284	10	3.5	1.0 (0.5–2.1)	0.866	2.1 (1.0–1.5)	0.208	1.0 (0.9–1.1)	0.563	1.7 (1.4–2.1)	<0.001
Southern–Eastern Europe	1030	13	1.3	0.3 (0.1–0.6)	0.002	0.4 (1.0–1.5)	0.176	0.4 (0.4–0.5)	<0.001	0.6 (0.5–0.7)	<0.001
North Africa and the Middle East	4218	230	5.4	1.7 (1.2–2.4)	0.003	1.6 (1.0–1.5)	0.071	1.5 (1.4–1.6)	<0.001	1.3 (1.2–1.5)	<0.001
South–East Asia	4940	154	3.1	0.9 (0.6–1.3)	0.742	0.8 (1.0–1.5)	0.655	0.9 (0.9–1.0)	0.202	0.8 (0.8–0.9)	0.014
North–East Asia	2540	74	2.9	0.8 (0.5–1.3)	0.530	1.5 (1.0–1.5)	0.194	0.9 (0.8–0.9)	0.008	1.2 (1.1–1.3)	<0.001
Southern and Central Asia	4011	168	4.2	1.3 (0.9–1.8)	0.146	1.6 (1.0–1.5)	0.088	1.2 (1.1–1.3)	<0.001	1.4 (1.3–1.6)	<0.001
Americas	501	7	1.4	0.4 (0.1–0.9)	0.035	0.3 (1.0–1.5)	0.188	0.5 (0.4–0.6)	<0.001	0.5 (0.4–0.6)	<0.001
Sub–Saharan Africa	675	29	4.3	1.3 (0.7–2.1)	0.285	1.3 (1.0–1.5)	0.526	1.2 (1.1–1.3)	<0.001	1.2 (1.0–1.3)	0.005
Obstetric factors											
Alcohol use in pregnancy											
No	17,001	619	3.6	1.0		1.0		1.0		1.0	
Yes	143	3	2.1	0.5 (0.1–1.8)	0.316	0.9 (0.7–2.1)	0.953	0.7 (0.2–2.0)	0.572	0.7 (0.1–3.1)	0.689
Antenatal health problems											
No	15,755	552	3.5	1.0		1.0		1.0		1.0	
Yes	3041	140	4.6	1.3 (1.1–1.6)	0.003	0.9 (0.6–1.3)	0.726	1.2 (1.0–1.5)	0.017	0.9 (0.8–0.9)	0.008
Type of delivery											
Normal vaginal	11,336	376	3.3	1.0		1.0		1.0		1.0	
Assisted vaginal	2323	84	3.6	1.1 (0.8–1.4)	0.351	1.6 (1.0–2.5)	0.032	1.0 (0.8–1.3)	0.501	1.4 (0.9–2.2)	0.096
Caesarean section	5658	262	4.6	1.4 (1.2–1.7)	<0.001	1.4 (1.1–1.9)	0.007	1.3 (1.1–1.5)	<0.001	1.3 (1.0–1.7)	0.034

EPDS: Edinburgh Postnatal Depression Scale; OOHC: out of home care; FACS: Family and Community Services; IPV: intimate partner violence; ^#^ Multivariate models adjusted for the birthing facility and gender of the baby, as well as relevant psychosocial and obstetric factors. * Sensitivity analyses following multiple imputations for missing values; OR (95% CI): Odds ratio with 95% confidence interval.

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
