# Peer review of "Perinatal Distress and Depression in Culturally and Linguistically Diverse (CALD) Australian Women: The Role of Psychosocial and Obstetric Factors"

_ijerph, 2019, doi:10.3390/ijerph16162945_

Round 1
Reviewer 1 Report
Review: Perinatal distress and depression in culturally and linguistically diverse (CALD) Australian women: the role of psychosocial and obstetric factors
Title: Ok
Abstract: ok
Keywords: should be in alphabetical order
Introduction: CALS should be abbreviated once
M&M: IPV should be explained in the first use
-what is SES?
- WHJAT IS epds?
-what is IM&TD ?
- what is CFHN
- what is MCH data?
- what was realized the need of anonymity of participants?
Results
Pleas clearly explain what is the control (mark control)
Discussion
It should be explained what is the problem if control used for comparison might indicate different results and sometimes might not be comparable
Author Response
Review: Perinatal distress and depression in culturally and linguistically diverse (CALD) Australian women: the role of psychosocial and obstetric factors
Title: Ok
Abstract: ok
Response:
We thank the reviewer for the comment.
Keywords: should be in alphabetical order
Response:
Keywords now revised in alphabetical order.
Introduction: CALS should be abbreviated once
Response:
Points appreciated and now reflected in the revised manuscript.
M&M: IPV should be explained in the first use
Response:
Points appreciated and now reflected in the revised manuscript.
-what is SES?
Response:
Points appreciated and the text has been edited in the revised manuscript, consistent with reviewer 2 comment, below.
- WHJAT IS epds?
Response:
EPDS means Edinburgh Postnatal Depression Scale. This was defined in the introduction section and is the most widely used tool in mental health screening for depression globally.
-what is IM&TD ?
Response:
IM&TD means Information Management and Technology Division. The text has been edited in the revised manuscript, consistent with reviewer 2 comment, below.
- what is CFHN
Response:
CFHN means child and family health nurses. The text has been edited in the revised manuscript, consistent with reviewer 2 comment, below.
- what is MCH data?
Response:
MCH means maternal and child health. The text has been edited in the revised manuscript, consistent with reviewer 2 comment, below.
- what was realized the need of anonymity of participants?
Response:
Participants anonymity was necessary as the overall study project does not seek to follow up respondents over time as noted in the ethics protocol submission.
Results
Pleas clearly explain what is the control (mark control)
Response:
The study used linked retrospective maternal and child health data (secondary information) of all live births in public health facilities among the population of CALD women in Sydney Local Health District (SLHD) and South Western Sydney Local Health District (SWSLHD) between 2014 and 2016 (N=25,407). Respondents to specific questions (e.g., physical intimate partner violence, IPV) were categorised into “Yes” or “No”. Those who reported “Yes” to physical IPV were cases, while those who reported “No” were controls.
This information was noted in the methods section of the original manuscript.
Discussion
It should be explained what is the problem if control used for comparison might indicate different results and sometimes might not be comparable.
Response:
We acknowledged as part of the study limitation, the issues relating to the data used.
Third, in the present study, the EPDS was based on self-report in the seven days prior to contact with the health practitioner in the perinatal period, potentially leading to recall and/or measurement bias. This may have resulted in an overestimation or underestimation of the association between the study factors and the outcomes. Fourth, we were unable to differentiate women who have a prior history of mental illness (such as anxiety and/or depression) in the perinatal period. This analysis would have provided detailed information on the role of the study factors on the outcomes among CALD Australian women. Fifth, the study was unable to assess or adjust for other potential determining factors (e.g., prematurity, level of support received during pregnancy and postnatally or multi-parity) as this may also affect the observed estimates. Sixth, the small sample size available for some of the study factors may account for the large effect sizes and the wide CIs. Seventh, the non-use of maternal and child health data from private healthcare and other Local Health Districts in Sydney is a limitation in the present study, as it affects the external validity. Finally, the unavailability of longitudinal data from conception to 12-months post-birth and the use of secondary data were additional limitations in this study.
Additionally, studies relating to breastfeeding and depression have previously been published using similar data.
References
Ogbo, F. A., Ezeh, O. K., Khanlari, S., Naz, S., Senanayake, P., Ahmed, K. Y., ... & Ussher, J. (2019). Determinants of Exclusive Breastfeeding Cessation in the Early Postnatal Period among Culturally and Linguistically Diverse (CALD) Australian Mothers. Nutrients, 11(7), 1611. Ogbo, F. A., Eastwood, J., Hendry, A., Jalaludin, B., Agho, K. E., Barnett, B., & Page, A. (2018). Determinants of antenatal depression and postnatal depression in Australia. BMC psychiatry, 18(1), 49. Ogbo, F. A., Eastwood, J., Page, A., Arora, A., McKenzie, A., Jalaludin, B., ... & Chaves, K. (2016). Prevalence and determinants of cessation of exclusive breastfeeding in the early postnatal period in Sydney, Australia. International breastfeeding journal, 12(1), 16. Eastwood, J., Ogbo, F. A., Hendry, A., Noble, J., Page, A., & Early Years Research Group. (2017). The impact of antenatal depression on perinatal outcomes in Australian women. PLoS One, 12(1), e0169907.
Reviewer 2 Report
The authors present a study of the risk factors related to perinatal distress and depression in culturally and linguistically diverse women in Australia. This is an interesting study in an understudied population.
The subject matter is important. Though the risk factors themselves don't appear to be novel, the population studied is more so. The discussion of policy implications of the data is nice.
Please provide the proportion of individuals identified as CALD in Australia.
Please provide more detail on the definition of socioeconomic status. It would be nice to know income ranges and mean to see how this might generalize to other studies.
The tables are informative but quite large. It might be a good idea to add an additional figure that summarizes key findings.
Perhaps too much usage of initialisms. CALD makes sense as it's wordy and the focus of the entire paper, so reader's will know what you're talking about. DALY is defined once and used only once more in the paper. Socioeconomic status is only two words, it seems like little is gained by spelling it as SES. Perhaps word count was a concern, but the manuscript might be more readable if fewer initialisms were used throughout.
Author Response
Comments and Suggestions for Authors
The authors present a study of the risk factors related to perinatal distress and depression in culturally and linguistically diverse women in Australia. This is an interesting study in an understudied population. The subject matter is important. Though the risk factors themselves don't appear to be novel, the population studied is more so. The discussion of policy implications of the data is nice.
Response:
We thank the reviewer for the comment and compliment.
Please provide the proportion of individuals identified as CALD in Australia.
Response:
Relevant information relating to the CALD population in Australia along useful references have been incorporated into the revised manuscript as requested by the reviewer.
The 2016 Australian Bureau of Statistics, Census of Population and Housing indicated that almost half of Australians (45%) were either born overseas or had one or both parents who were born overseas (19%). In Sydney, the SLHD and SWSLHD cover 52% of the metropolitan area, with an estimated population of 1.6 million people of different cultural backgrounds [33, 34]. In the Sydney metropolitan area, more than half of the population spoke English at home (58.4%). Other most common languages spoken at home included Mandarin (4.7%), Arabic (4.0%) and Cantonese (2.9%). [35].SLHD is located in the centre and inner west of Sydney, while SWSLHD is located in the south-western region of the city. A number of maternal and child health services are provided to all communities across both districts, including those with socioeconomically disadvantaged populations [33, 34].
Please provide more detail on the definition of socioeconomic status. It would be nice to know income ranges and mean to see how this might generalize to other studies.
Response:
Additional text has been incorporated into the revised as requested by the reviewer (Page 4, Paragraph 1).
Socioeconomic status was calculated using the Socio-Economic Index for Areas (SEIFA). SEIFA is an indicator created by the Australian Bureau of Statistics using principal component analysis and is an area-based scale in Australia according to socio-economic advantage and disadvantage. The variables used in the estimation of SEIFA cover a number of areas, including household income, employment, education, occupation and housing, as well as other indicators of advantage and disadvantage [37]. In the present study, deciles of socioeconomic status were categorised into high (top 10% of the population), middle (middle 80%) and low (bottom 10%) groups, similar to previously published studies [6, 21].
The tables are informative but quite large. It might be a good idea to add an additional figure that summarizes key findings.
Response:
We agree with the reviewer that the tables are informative. We, however, note that any additional summary figure would just be a repeat of the information in the tables and results section especially this section of the manuscript has summarised the key findings requested by the reviewer. Additionally, any extra figure would be adding to the already ‘large’ tables in the manuscript. Most importantly, our presentation of the results alongside the tables is consistent with contemporary reporting of epidemiological studies.
Perhaps too much usage of initialisms. CALD makes sense as it's wordy and the focus of the entire paper, so readers will know what you're talking about. DALY is defined once and used only once more in the paper. Socioeconomic status is only two words, it seems like little is gained by spelling it as SES. Perhaps word count was a concern, but the manuscript might be more readable if fewer initialisms were used throughout.
Response:
Points appreciated, and relevant abbreviations have been edited in the entire revised manuscript.
